# Bio-Based Production Systems: Why Environmental Assessment Needs to Include Supporting Systems

**Andreas Nicolaidis Lindqvist [1,2,]*** [ID], **Sarah Broberg [1], Linda Tufvesson [2], Sammar Khalil [2] and Thomas Prade [2]** [ID]

1   RISE Research Institutes of Sweden, Ideon Beta5, Scheelevägen 17, 22370 Lund, Sweden
2   Swedish University of Agricultural Sciences, Department of Biosystems and Technology, P.O. Box 103, SE-230 53 Alnarp, Sweden
*   Correspondence: andreas.nicolaidis@ri.se

**Abstract:** The transition to a bio-based economy is expected to deliver substantial environmental and economic benefits. However, bio-based production systems still come with significant environmental challenges, and there is a need for assessment methods that are adapted for the specific characteristics of these systems. In this review, we investigated how the environmental aspects of bio-based production systems differ from those of non-renewable systems, what requirements these differences impose when assessing their sustainability, and to what extent mainstream assessment methods fulfil these requirements. One unique characteristic of bio-based production is the need to maintain the regenerative capacity of the system. The necessary conditions for maintaining regenerative capacity are often provided through direct or indirect interactions between the production system and surrounding "supporting" systems. Thus, in the environmental assessment, impact categories affected in both the primary production system and the supporting systems need to be included, and impact models tailored to the specific context of the study should be used. Development in this direction requires efforts to broaden the system boundaries of conventional environmental assessments, to increase the level of spatial and temporal differentiation, and to improve our understanding of how local uniqueness and temporal dynamics affect the performance of the investigated system.

**Keywords:** bioeconomy; bio-based economy; bio-based production systems; environmental assessment; sustainability assessment; LCA; environmental management; systems analysis

## 1. Introduction

Transitioning to a bioeconomy (or bio-based economy) is a high political priority on both the national and the European level. According to the strategy and action plan of the European Commission, *Innovating for sustainable growth: a bioeconomy for Europe* [1], and the subsequent *Updated Bioeconomy Strategy* [2], the bioeconomy "*encompasses the production of renewable biological resources and the conversion of these resources and waste streams into value-added products, such as food, feed, bio-based products, and bioenergy*". The objective of the transformation is to achieve sustainable development by tackling several societal challenges simultaneously, e.g., ensuring food security, sustainable management of resources, replacing non-renewable resources with renewables, mitigation of and adaptation to climate change, job creation, and maintaining economic competitiveness [3]. In addition to the EU strategy, several countries are currently developing, or have developed, their own bioeconomy strategies, including Sweden, the Netherlands, the US, Malaysia, South Africa, Germany, Finland, and France [4,5]. Even though there are variations in definitions and wording, and there are differences in aims and objectives, driving forces, sustainability perspectives, spatial focus, and in the role of technology innovations (see for example Bugge, et al. [6]), there are certain key characteristics common to most

bioeconomy strategies. First, the transition to a bioeconomy calls for the increased production and extraction of biomass. The biomass should be utilized to provide food and feed, as well as broadly replace non-renewable resources across sectors, including the transportation sector, the energy sector, chemical industries, construction, life sciences, etc. Second, enabling the widespread replacement of non-renewable resources by renewables requires research and the commercialization of "green technologies", such as biomass processing, biotechnology, and biorefinery concepts. With innovative technologies, biomass has the potential to be a substitute to oil, gas, and coal in most of their current applications, and thereby reduce our reliance on fossil resources. Third, resource efficiency should be achieved through a cascading use of resources, the valorization of residuals, and the adoption of circular economy principles. Fourth, the transformation towards a circular bioeconomy, and the application of green technologies, is part of the solution to several of the sustainability challenges facing society today (including climate change, ecosystem degradation, resource depletion, biodiversity loss, food insecurity, etc.). Fifth, by expanding the market for bio-based products, transforming the industrial sector, fostering biotechnological innovation and supporting rural development, the bioeconomy enhances economic growth and creates new jobs across all sectors of society [1,4–7].

Ensuring that the multiple sustainability objectives of the bioeconomy transformation are achieved is a significant and challenging task. In this study, we have therefore narrowed our scope by directing our efforts to the specific challenges associated with assessing the environmental dimension of sustainability (from here on referred to simply as sustainability) in bio-based production. Such an assessment requires methods that are both comprehensive and profound, as well as being adapted to the particular characteristics and inherent complexity of the bio-based production systems [8]. Life Cycle Assessment (LCA) has often been described as the key assessment and planning tool for this purpose [3]. LCA is one of the most commonly used tools for environmental assessment, it has been applied for environmental assessment and planning in multiple sectors globally, and it is widely used as a decision support tool for product development, environmental benchmarking, management, and policy development [3,9]. However, despite its popularity, researchers have raised concerns regarding some of the limitations and weaknesses of the LCA methodology, and worries have been expressed that, if not addressed, these limitations and weaknesses can have significant implications for the applicability and reliability of the assessments [9–16]. For example, Reap et al. [12,17], conducted an extensive literature review on unresolved problems related to LCA methods and application, and identified multiple problem areas, with potentially significant implications in terms of reliability and usefulness as a sustainability assessment tool [12,17]. Furthermore, in light of the current bioeconomy discourse, Cristobal et al. [18] state that current limitations in the LCA methodology severely limit our understanding of the environmental implications of bioeconomy value chains, and this constitutes a significant problem for management and policy development. Thus, improving our capacity to assess the environmental impacts of bioeconomy development is of great importance for ensuring the sustainability of the transition at hand.

This need for robust assessment methods, adapted for the specific needs of bio-based systems, and the expressed concerns regarding the inherent limitations of the LCA methodology, constitute the starting point for this study. The aim was to investigate: (1) What are the key characteristics of bio-based production systems that need to be taken into account when assessing their long-term sustainability? (2) How do these affect the suitability and reliability of LCA as the primary assessment and planning tool for the bioeconomy? A third aim was to (3) provide guidance and direction for future research, in terms of important aspects to consider in future assessment and planning of bio-based production systems. Achieving these aims requires both a comprehensive understanding of how bio-based production systems differs from the current, largely fossil based, economic discourse, and how these differences influence the requirements of the sustainability assessment. To this end, the objectives of this paper are: (a) to outline environmental aspects that are of specific importance to address in sustainability assessments of bio-based production systems, compared to systems based on non-renewable resources; (b) to explore the requirements these aspects impose on the sustainability

assessment, and to what extent the current LCA methodology fulfils these requirements, and; (c) to provide recommendations on areas to improve in future assessment and planning efforts of bio-based production systems, based on the identified aspects and requirements.

## 2. Materials and Methods

To answer the first two questions above, we conducted extensive literature studies on the bioeconomy concept and the debate regarding its environmental and ecological sustainability, environmental aspects of bio-based production systems, natural resource management theory, and the role of LCA in the bioeconomy transition. Additionally, a literature review was conducted focusing on the limitations of the LCA methodology when applied to bio-based production systems. The review included published scientific papers (both reviews and articles), books on LCA methodology, reports, and governmental publications. The primary source used for scientific publications was Web of Science. Google Scholar and Google were also utilized for retrieving "grey literature", such as reports, government documents or publications by non-governmental organizations, and for finding references not covered by the Web of Science databases. The scope of the literature search in Web of Science was limited to publications between 2008 and 2018, and further limited by using topic-based searches combining the terms "Sustainability assessment" AND bioeconomy, "Sustainability assessment" AND "bio-based economy", "Sustainability assessment" AND "bio-based system*", LCA AND Weakness*, LCA AND Limitation*, LCA AND "Research need*". The broad choice of search terms was intentionally used to ensure that relevant studies without an explicitly stated bioeconomy/bio-based focus were not excluded. No geographical restrictions were applied, and only studies published in English or Swedish were considered. With these search criteria, 616 publications were found. Additionally, to broaden the scope and add information from other sustainability assessment methods, a reviews-only search was conducted with the terms "sustainability assessment" AND methods, yielding 88 additional publications, giving a total of 704 scientific papers. Titles and abstracts were scanned, and selection criteria for identifying relevant articles, based on the above stated aims and objectives, were applied: (1) to focus on limitations and problem areas in LCA and other methods for assessing environmental sustainability; (2) to focus on bioeconomy and/or bio-based production systems; and (3) to focus on the need for, and approaches to, improving the environmental assessment methodology as a tool for the assessment and planning of bio-based production systems. Articles considered relevant to one or more of the criteria were selected and read in detail. When appropriate, key references of selected articles were also retrieved and included in the literature review. The selection process was carried out by the review first author, however, to ensure the appropriateness of search criteria, selection criteria, coverage, interpretation of data, etc. The full group of authors was regularly consulted throughout the process. In total, 107 scientific articles and review papers, and 28 books, book sections, reports, and other "grey literature" sources were included in the review.

The selected literature was qualitatively analyzed and the results are presented in the chapters below. First, we analyzed the documented key differences between bio-based production systems and systems based on non-renewable resources and explored the requirements for bio-based production systems to be considered sustainable (Section 3.1). Second, we investigated what the characteristics and requirements of bio-based production systems mean for the sustainability assessment in terms of scope, system boundaries and choice of impact categories. We identified impact categories documented as particularly important for the assessment of bio-based production systems, and we explored to what degree these are covered my mainstream assessment methods (Section 3.2). Third, we evaluated what challenges the identified impact categories impose on the sustainability assessment and to what extent current assessment methodology addresses these challenges (Section 3.3). In chapter four, we discuss three areas in need for targeted efforts to address the methodological challenges associated with sustainability assessment of bio-based production systems and, in chapter five, we provide our own reflections and recommendations for future researchers and practitioners to keep in mind, in order to improve the environmental sustainability assessment of bio-based production systems.

## 3. Results

### 3.1. What Is a Sustainable Bioeconomy?

Even though environmental sustainability is at the core of the European bioeconomy strategy [2], a fossil-free economy, built upon bio-based production, the cascading use of resources, and advanced green technologies, is not sustainable by default. Biodiversity loss, ecosystem degradation, land-use-change, freshwater depletion, and greenhouse gas emissions are all examples of possible environmental impacts from unsustainably managed bio-based systems [4,19–21]. Therefore, planning and transitioning towards a sustainable bioeconomy calls for assessment methods that are tailored towards the specific environmental issues of bio-based production [19,22]. This, in turn, requires a comprehensive understanding of the specific characteristics of these systems (mode of operation, critical environmental aspects, etc.), and how their environmental issues and sustainability challenges differ from those of the current fossil-dependent discourse [18,19].

In this study, we define bio-based production systems as open, or semi-open, social–ecological systems that combine human technology and biological processes to utilize the ecosystems, their services and biological resources, for the production of food, fiber, biomass or other bio-based products [23]. The concept encompasses traditional cropping and animal systems for food and feed, forestry for timber and energy purposes, fisheries and aquaculture, as well as more novel systems for the production of biofuel and bio-chemicals (e.g., algae farming or bio-energy cropping systems) [23]. Bio-based systems are unique in their inherent capacity of regeneration, allowing biological resource stocks to replenish after extraction. In theory, biological resources can be continuously exploited for eternity as long as two fundamental conditions are met: (a) the rate of extraction does not exceed the rate of regeneration [24], and (b) the extraction, processing, and utilization of the resource, and other external factors, do not diminish its regenerative capacity of the system. If these two criteria are met, the resource can be considered renewable, which is a prerequisite for the system to be considered sustainable [19,24,25].

In contrast, production systems based on fossil resources, minerals, and metal ores, are non-renewable. This means that there is a finite amount of these resources available in the earth's crust and no regeneration occurs (or the regeneration rate and the processes involved in regeneration are so slow that they are neglectable from a human time perspective). Since fossil/non-renewable resources do not regenerate, these systems are not depending on the maintenance of a regenerative capacity. Therefore, in the sustainability assessment of fossil resource systems, greater emphasis should be on ensuring that waste emissions from extraction and utilization do not lead to the degradation of surrounding systems, and that the rate of extraction should not be faster than the rate of development of renewable substitutes to replace the fossil resource [24]. Assessments of bio-based production systems also need to focus on minimizing emissions and damage to surrounding systems, however it is the need to ensure that the regenerative capacity of the system is maintained that is the key difference that makes sustainable management of these systems fundamentally different from their non-renewable counterparts.

It is important to note that the regenerative capacity of biological resources is not static. On the contrary, it is tightly correlated with both the state of the resource stock itself and the state and availability of other limited resources (e.g., water, land, nutrients, soil or suitable habitats [19,22]). This entails that in order for condition (b) to be met, these critical resources need to be maintained within required limits to support regeneration.

Another important difference is that bio-based production systems are typically tightly connected, and in constant interaction, with their surrounding systems [26–28]. For example, a forest is in constant interaction with the surrounding atmospheric system through the exchange of $CO_2$ and oxygen [29], agricultural systems are tightly connected to, and affected by, the surrounding hydrological system [30], and many fisheries are influenced by the state of distant river and freshwater systems for spawning [31]. Very often, it is through these system interactions that the critical resources for regeneration are maintained within required limits. For example, the productivity and regenerative

capacity of an agricultural field is influenced by the capacity of the soil to replenish, retain water, and provide necessary plant nutrients; the surrounding hydrological system influences crop water availability; and pollination is influenced by the capacity and resilience of neighboring ecosystems. These interdependencies are often bilateral. The soil quality is affected by agricultural practices, such as biomass extraction and fertilizer use, and the hydrological cycle is influenced by irrigation practices, and how this change evapotranspiration and water retention time, etc. Thus, interactions with surrounding systems are constantly affecting the rate of regeneration in bio-based systems, and thereby influencing future resource extraction.

In contrast, fossil resources typically do not interact with surrounding systems (ecosystems, social systems, physical systems). This is either because the resource itself is inert (e.g., many metal ores are chemically unreactive), or, as for many petroleum resources, because of physical boundaries isolating the resource from its surroundings, e.g., oil reservoirs are typically confined by some geological formations (e.g., impermeable rock or salt). This physical confinement plays an important role in the chemical formation of the petroleum resource and, more importantly, it isolates the oil and gas from any interactions with surrounding systems. Even though the process of extracting, processing, and utilizing fossil resources often has significant environmental impacts, in the form of greenhouse gas emissions, land degradation, forest clearance, chemical pollution, etc. [32–34]. The change in the state of the fossil resource (the size or quality of the resource stock) does not profoundly influence the surrounding systems. For example, a deep-sea oil deposit does not interact with the surrounding marine ecosystems, with the marine food web, or with the fishing communities utilizing the surrounding waters. Thus, surrounding systems are not affected by changes in the size or state of the resource stock. The same is true in the other direction. Since the future extraction of fossil resources is not dependent on a maintained regenerative capacity, any changes in state of the surrounding systems have very limited influence on the future extraction, i.e., the eutrophication of surrounding waters does not affect the size or state of the oil deposit because the production/formation of the oil has no connection to the state of the surrounding systems.

In summary, a sustainable bioeconomy requires that bio-based production systems are managed so that the rate of extraction does not exceed the rate of regeneration, and that the regenerative capacity of the resource stock is maintained. For this to be possible, management must also consider the interactions between the biological resource stock and the surrounding "supporting" systems responsible for providing the necessary conditions for regeneration. This makes the management of bio-based production systems much more complex than the management of fossil-based systems. Fossil resources are typically systemically inactive, and management primarily needs to focus on minimizing environmental impact from extraction, processing, and utilization. Draining the fossil resource stock typically has no direct implications for surrounding systems, neither do surrounding systems influence the size of the stock, or change the conditions required for exploitation. Thus, exploitation and management of fossil and biological resources require fundamentally different strategies. The latter requires a comprehensive system perspective, where not only the size and state of the primary resource stock is maintained, but also the size and the state of surrounding systems involved in providing the necessary conditions for regeneration.

### 3.2. Assessing the Environmental Sustainabilityt of Bio-Based Production

Since the production and regenerative capacity of biological resources depend on both the state of the production system itself, and on the state of, and interactions with, neighboring systems, these must all be included in the sustainability assessment. In LCA, this means that impact categories should be chosen so that effects on both the production system and on supporting systems are included in the assessment. For example, in agricultural production, the soil system is one of the supporting systems providing the necessary conditions for biomass production and regeneration (providing plant nutrients, water retention, and growth substrate). Thus, in order to truly assess the sustainability of

the production system, the system boundaries of the study need to be sufficiently wide so that impact categories related to the state of important soil parameters are included in the assessment [35].

On a conceptual level, having broad system boundaries, and including multiple, parallel impact categories in the analysis is not a problem. In fact, one of the well-documented assets of the LCA methodology is its capacity to address multiple environmental issues simultaneously, and that this helps avoid burden shifting between environmental impacts, and across time and space [36,37]. Covering multiple impact categories should ensure that efforts for lowering one environmental impact does not unintentionally cause trade-offs with another one, e.g., reducing greenhouse gas emissions at the expense of increased eutrophication [37–39]. However, this is not often the case, as the number of impact categories considered is often restricted to a selected few [40–42]. Also, scanning the LCA literature shows that not only is the representation of impact categories often incomplete, but it is also often highly skewed. Some impact categories are overrepresented, and others are only rarely represented [36,43]. For example, Global Warming Potential (GWP) was included in 98% of livestock LCA studies reviewed by McClellande et al. [42] (from a total of 173 papers published between 2000 and 2016, 169 studies included climate change as an impact category) and in 97% of LCA studies on biofuels reviewed by Lazarevic and Martin [36]. Biodiversity and ecosystem services (ESs), on the contrary, were only covered in 3% of the studies reviewed by McClellande et al. [42], and in none of the ones investigated by Lazarevic and Martin [36]. Water resource depletion and biotic resource depletion were not included as separate impact categories in any of the papers reviewed by Lazarevic and Martin [36], nor by McClellande et al. [42]. Instead, these were incorporated as part of the impact category "resource depletion" (broadly including both biotic and abiotic resources). The often limited and uneven coverage of impact categories in the LCA literature can be explained by a number of factors. Limited time, budget, and data availability are common issues constraining the choice of categories [18]. Trends in politics, research, and media focus are other influential factors [44]. For some impact categories, the availability of quality data and the lack of well-established impact models are other bottlenecks. As examples, assessments of impacts on biodiversity, ESs, and water use often suffer from a lack of quality data and available impact models [9,12]. Thus, these are less likely to be included in a study, compared to other categories with less complex, or better documented, cause–effect chains [30,40,45–47]. The multifunctional nature of bio-based systems, and the system–system interactions they rely on, make adequate impact category coverage particularly important. For example, Lorilla et al. showed how the state of Mediterranean agricultural production systems affects the functioning of several ESs through complex system interactions [48] and in a study focusing on land-use impacts on soil quality parameters, Vida Legaz et al. presented intricate cause–effect relationships shaping the impact pathway, from changes in soil conditions to impacts on biomass production, freshwater provisioning, climate regulation, biodiversity, etc. [35]. Capturing these types of synergies and feedbacks, and ensuring they are covered in the impact assessment, is a challenge in LCA [13,26].

Given time and funding restrictions, limiting the number of impact categories is often the only option available [49], and it is sometimes legitimized as a way of reducing the complexity of the study and providing a clearer message to the audience [18]. However, the uneven coverage of environmental impacts can be problematic for the credibility and usability of LCA results, as it increases the risk of problem shifting [36,42]. It can also give the impression that some environmental issues are non-existent when, in reality, they have simply not been covered by the analysis. This was demonstrated by Berger, et al. [50] in a study comparing the water and carbon footprints of biofuels with those of fossil fuels. The results showed that, if focusing only on carbon footprint, biofuels perform better than fossil fuels due to their relatively lower net $CO^2$ emissions. However, when adding water footprint, and impacts on freshwater reserves, the sustainability of biofuel production was in many cases less obvious. Thus, impact category choice needs to be justified at an early stage of the assessment [51], and the choice should be tailored to the system studied. For bio-based systems, this entails including necessary impact categories to assess impacts affecting the stock of the resource and its regenerative capacity. Our analysis suggests that there are four impact categories that are particularly important for this purpose:

biotic resource depletion; freshwater use; biodiversity loss; and the degradation of ESs [20,26,52]. Interestingly, these impact categories are also among the least represented in the LCA literature, and several researchers have expressed the need to develop the assessment methodology to better account for these environmental issues [15,42,43,52,53].

### 3.3. Implications for LCA

Thus far, we have concluded that the sustainable management of bio-based systems requires different strategies compared to systems based on non-renewables, and that some of the most important impact categories to consider in the assessment of these systems are: biotic resource depletion; freshwater use; biodiversity loss; and impacts on ESs. We have also seen that these impact categories are underrepresented in the LCA literature, and partially this is because of limited data availability, and the lack of reliable models describing the effects bio-based systems may have on these impact categories. Due to these limitations, studies with restrictions in terms of time and budget tend to prioritize other impact categories, where data are more easily accessible and impact pathways more well-documented. Next, we will investigate in more detail why these impact categories are so challenging to assess, what requirements they impose on the sustainability assessment, and to what extent current LCA methodology can fulfil these requirements.

### 3.3.1. Biotic Resource Depletion

"Biotic resources" is a broad concept, encompassing a wide array of biological products and capital, including fish, wood, soil, etc. [54]. Historically, biotic resources have received limited attention in LCA and, in assessments of production systems based on biotic resources, the impacts from the depletion of the resources themselves are not accounted for in most cases [43,55]. The reasons are, in part, because of the lack of reliable indicators for many biotic resources, limited understanding of their associated elementary flows, and missing impact models that account for impacts on both the resource stock itself and indirect impacts on surrounding systems [37,43]. It is only in the last few years, with the growth of the bioeconomy, that the criticality and need for improvements in the impact assessment of biotic resource use have started to become recognized [1,43,55]. Yet, to date, these resources remain poorly addressed in LCA research (e.g., top-soil, forest biomass, and fish stocks) [18,43,56], and there is a lack of consensus on methods for assessing the system level impacts of their exploitation [43,54,57].

One major obstacle is the broad scope of the biotic resource concept, and how to cover the many different types of resources it encompasses. Currently, the coverage of different biotic resources in LCA inventories is far from complete, and the level of aggregation is typically high. For instance, wood biomass, a highly versatile biotic resource, is being harvested from forests across the globe, originating from different tree species, and different ecosystems and habitats (managed and natural). However, in LCA inventories, these different flows of wood biomass are typically referred to as simply "wood", or at best, a distinction is made between "softwood" and "hardwood" [43]. Looking at biotic resources covered by established LCA databases (e.g., Ecoinvent 3.3 [58] and the European Reference Life Cycle Database, ELCD) [59]) confirms that this is not a problem unique for wood, but for most biotic resources. For example, in the ELCD inventory database, "biomass" is represented as a single aggregated category, without any distinction between different types of biomass, its origin, or what species it originates from. Similarly, the Ecoinvent inventory databases aggregate marine fish into a single resource flow, regardless of population or species. This level of aggregation is problematic, as it does not distinguish between biotic resources of the same category taken from different species or habitats (e.g., wood sourced from cosmopolitan vs. endemic species), and it does not account for important ecological aspects, such as variations in species vulnerability, species resilience, minimum viable population size, regeneration rate, etc. With the conditions for sustainable bio-based production systems in mind, this lack of information on ecological characteristics makes it impossible to develop reliable impact models for these resources, and to assess the system-level impacts from their exploitation [43]. Thus, there is

an urgent need to improve the currently limited coverage of biotic resources in established life cycle impact (LCI) databases. Particularly urgent, in light of their role for the global economy, are categories such as topsoil, forest biomass, and commercial fish stocks [60].

The next challenge, after increasing the coverage of biotic resources in the assessment, is that of assessing the environmental impacts of their exploitation. To this end, indicator choice will have major implications and, currently, mass accounting is by far the most commonly used method in life cycle impact assessment (LCIA). On the one hand, accounting based on mass is simple and straightforward, extraction/harvest data are often readily available, and direct effects on the resource stock can be easily calculated. However, the approach has important limitations. For instance, characterization of environmental impacts based solely on mass is not straightforward, as the magnitude of impact will depend on the state of the resource stock and its supporting systems [43]. For example, the impact from harvesting 1000 tons of fish from a fish stock that is close to its Biomass for Maximum Sustainable Yield (BMSY) will be very different when compared to the same amount being harvested from a stock that is significantly below its BMSY [61]. In terms of mass, the impact is the same but, looking at regenerative capacity, the latter fish stock is likely to require a substantially longer time to recover from the extraction, because a lower regeneration rate is strongly correlated with population size. Another challenge is in how to account for quality aspects of the resource stock. For instance, the level of genetic diversity within the population will affect recovery rate after harvest, as well as how resilient the reduced stock is to external shocks. Recovery rate and resilience are likely to be greater if the genetic diversity within the population is high, compared to a population where genetic diversity is low [62,63].

Topsoil is another biotic resource in great need of better integration into the LCA methodology. Topsoil can be considered a primary biotic resource that can be depleted both by physical removal (e.g., through erosion or direct human interventions), or by quality degradation (e.g., the depletion of soil nutrients, changes in soil structure, salinization, etc.) [35]. However, similar to wood, the heterogeneity and variability in soil types and soil quality across the globe is significant, and different production systems have different soil requirements (e.g., pH requirements differ between coniferous and broad-leaved tree species). Therefore, assessing environmental impacts from soil degradation requires a comprehensive assessment methodology that takes this variability into account, rather than assuming impacts to be homogenous for different systems in different settings [35]. Further, the soil system is also one of the very important supporting systems involved in maintaining the regenerative capacity of many bio-based production systems. For instance, soil quality and soil productivity significantly affect forest regeneration [64,65], and studies in Poland have shown how variability in soil type, within the same forest, can increase or decrease tree recruitment by up to 300% [66]. Soil quality factors also have a significant impact on agricultural resilience and productivity (for some crops, the correlation coefficient between soil quality and yield has been documented to be as high as 0.9) [67,68].

According to the LCA standards provided by the International Organization for Standardization (ISO), the impact categories chosen in a given study should comprehensively cover environmental issues related to the targeted production system, while taking the goal and scope of the study into consideration [69]. However, as has been presented above, many environmental issues related to system-level impacts, and implications for the longevity and regenerative capacity of biotic resources, are currently not captured by the mainstream LCA methodology. This is alarming, as these constitute factors of great importance for making informed decisions regarding future biotic resource management [43,70].

Among existing efforts to overcome this gap, Langlois et al. presented alternative methods for LCIA of biotic resource depletion in fisheries where the Maximum Sustainable Yield (MSY) concept was incorporated into the impact assessment methodology, together with ecological aspects such as estimations of regeneration capacity [61]. Similar attempts have been made for terrestrial biotic resources [50,71], and Crenna et al. [43] presented an innovative approach where the renewal rate was calculated for a number of natural biotic resources (ranging from terrestrial to aquatic, and from

mammals to algae), measured in years required for reproducing one kilogram of the resource after extraction. In general, however, a higher degree of case specificity (ecological features, local conditions, socio-economic structures, etc.) is needed for these approaches to be successful, as they currently do not take into consideration the significant variability in renewability rates—governed by the state of the resource and its interactions with the surroundings [43]. Schneider et al. [54] takes this further, suggesting that even the specific extraction site, and its surroundings, need to be explicitly modelled for an adequate assessment of impacts on biodiversity and ecosystems resulting from biotic resource depletion.

To summarize, assessing the environmental impacts from biotic resource use requires an improved coverage of different biotic resources than what is currently the case. Due to significant differences in renewal rate, geographical distribution, resilience to shocks, etc., between and within biotic resources, the current practice of aggregating these into broad categories, such as "wood" or "fish", makes impact assessment difficult. To really capture the environmental impacts from biotic resource depletion, the specific characteristics of the resource studied needs to be considered, and impacts should be studied from both a "resource perspective", focusing on the state of the resource stock itself, and from a "system perspective", focusing on environmental impacts caused through the interactions and interdependencies between the resource stock and its surrounding systems.

### 3.3.2. Freshwater Use

Freshwater is a key resource in terrestrial bio-based production systems, and a medium for different types of aquatic production (e.g., freshwater aquaculture and freshwater fisheries) [20]. As with biotic resource use, freshwater use has historically received limited attention in LCA [72]. Most existing impact assessment methods primarily use a volumetric approach [15], focusing on the volume of water extracted from a watershed by a studied activity over a given period of time. The result has often been that regions with a history of abundant water supplies have gradually disappeared from the environmental water debate [73]. This is unfortunate for several reasons. Firstly, freshwater supplies are dynamic and constantly changing and, hence, historically abundant supplies are not a guarantee of future water access. Furthermore, freshwater systems are complex and interconnected structures, often stretching over large geographical areas, and thus water extraction in abundant parts of the watershed can influence the water supply in other areas further downstream in the system [73]. Another important aspect is that freshwater is both an abiotic resource and an environmental compartment, and processes altering the hydrological compartment in one end of a watershed can have serious implications for areas further downstream. For instance, water-polluting substances can be released in low concentrations in one part of the system, without any detrimental impact on water quality, but cause degradation in water quality and restricted water access for distant users, as the pollutants accumulate over time further downstream in the same watershed (e.g., by rendering the water system unsuitable for aquaculture purposes). Based on these characteristics, assessments of the environmental impacts of freshwater use need to take at least three types of usage into account, consumptive use, degradative use, and in-stream use, and do so with indirect upstream and downstream impacts in mind [30,74]. In this review, the focus is on consumptive and in-stream use, as degradative use is typically covered by impact categories related to pollution (e.g., eutrophication and freshwater toxicity) [30].

Most LCIA methods focus on consumptive water use (that is, water that is withdrawn and not released back into its original source) [30]. Water To Availability (WTA) [72], Distance To Target [60], and the Water Stress Index (WSI) [74] are examples of approaches for assessing water consumption in LCIA that attempt to do so by incorporating relative freshwater availability in the impact assessment models. Even though much work has been done in developing these methods, several limitations still exist. Firstly, these methods typically are concerned with water withdrawal required for human activities, and limited attention is given to ecosystem needs. For bio-based production systems, this means that the assessment does not account for how the water consumed by the production system

impacts the regeneration rate of the biological resource, nor the impacts on the supporting ecological systems involved in maintaining regeneration. For instance, studies have shown how water stress reduces pollination services in agriculture systems by limiting nectar production, flower development and by reducing habitat suitability for pollinators, thereby undermining the regenerative capacity of the system [75–77]. Canals, et al. [78] and Smakhtin, et al. [79] provided notable exceptions, as their methods not only accounts for human water needs, but also for environmental freshwater requirements (EFR). This is done by including impact pathways between freshwater consumption and the effects on surrounding ecosystems, e.g., by accounting for effects of changes in water availability for aquatic ecosystems, or effects on wetland habitats from lowered groundwater tables. Another limitation, when assessing water consumption, is that data on local water use and availability are often limited, and researchers often need to rely on regional average values, or extrapolate data from previous studies [30]. This typically causes high levels of uncertainty, as contextual and temporal variations in water availability and water needs are not fully captured [30,80]. For instance, studies have shown that river ecosystems can be highly sensitive to periodic droughts, and to the alteration of natural flow regimes caused by temporal peaks in water consumption [81]. Using yearly average values of water use when assessing ecosystem impacts from these activities evens out any inter-annual variations in the water withdrawal and water availability, thus masking potential alterations of the flow regime, and subsequent impacts on the ecosystem. It seems that really assessing the ecosystem impacts from freshwater use requires considerable knowledge regarding the spatial and temporal dynamics of the water extraction, as well as detailed information on the ecosystem's composition and hydrology [82]. In many cases, obtaining information at this level of detail is costly and resource demanding. Thus, a higher contextual resolution of the assessment needs to be balanced against the added costs and effort that this entails. If data collection and model development are too costly or time-consuming, the likelihood of application remains very low despite the potential improvements in model fidelity, and the subsequent quality of the study.

In contrast to the consumptive use, environmental impacts of in-stream water use, and the subsequent alteration of flow regimes, are rarely addressed in LCA studies, even though human structures and activities are known to affect hydrological systems, water resource availability, and ecosystem functions [30]. For example, water regulation, or drainage of wetlands for agricultural purposes, is a common phenomenon known to have affected more than 65% of natural wetlands in Europe and North America [75]. In the short term, wetland drainage might increase agriculture productivity by expanding the land available for agriculture. However, drainage also reduces many important regulating ESs, causing unintended outcomes, and hitting back on agricultural productivity, e.g., by increasing the vulnerability of the agricultural system to extreme weather, and by increased soil and nutrient runoff [75]—undermining the capacity of the system to regenerate. In this review, very few methods for assessing in-stream water use by LCA were identified. Two examples are Humbert and Maendly [83], who developed characterization factors for assessing impacts on aquatic biodiversity from hydropower production, and Gracey and Verones [84], who investigated the effects of hydropower production on aquatic and terrestrial biodiversity. Hydropower production was shown to have significant negative effects on biodiversity and ESs via its impacts on hydrological flows, geomorphology, water quality, and habitat fragmentation [83,84]. However, for many of these impact pathways, the assessment methodology is still not fully developed, or even non-existent, and thus there is a great need for further research and method development [84].

The state of freshwater resources can be a fundamental constraint or facilitator to bioeconomic growth and bio-based production, and efficient management is therefore of great importance [20]. However, the number of studies investigating how the current and future state of freshwater resources may influence the growth of the bioeconomy are few. Exceptions include Rosegrant et al. [20], who investigated future scenarios of how water scarcity might influence food production and food security, and concluded that the effect of bioeconomy development on future water availability and food security will depend on multiple factors. Technology development and adoption, crop selection, historical

water-use efficiency, and governance structures for water management are all examples of factors in bioeconomy development that can be detrimental to future water availability. Berger et al. [50] conducted a study on the potential sustainability trade-offs between water use and the carbon footprint of European biofuels, and Ercin and Hoekstra [85] did a similar study on animal products. Furthermore, Veldkamp et al. [86] studied global data from the period 1971–2010, and concluded that human water interventions (land-use and land-cover changes, reservoir constructions, and water consumption) have historically contributed to changes in the geographical distribution of water-stressed regions, as well as to alterations in the dimensions of water scarcity in several of the studied areas. For most river basins, human interventions had an alleviating effect on water stress in the area of implementation, but an aggravating effect for areas further downstream from the intervention (increasing the level of stress for already water-scarce regions, or even pushing some areas into water stress). The overall trend observed, on regional and global levels, was that human interventions historically have caused water stress to travel downstream from the river basin [86]. These studies, the discussion above, and the documented high water use in many bio-based production systems, highlight the importance of thoroughly assessing the potential impacts the bioeconomy may have on freshwater scarcity. It is a possibility that bioeconomy development may become a driver of freshwater scarcity in some regions, and that water availability may become a limiting factor to bioeconomy development in other regions. Unless these impacts are carefully considered in the management of bio-based production systems, these systems may well undermine the long-term sustainability of the bioeconomy and contribute to water conflicts across regions.

Our findings highlight the importance of spatial differentiation and contextualization when assessing water use, and when translating it into impacts on the environment and the sustainability of bio-based production systems [50]. Several other studies support the need for more contextualized and dynamic impact assessment models as a complement to LCA, and the use of scenario analysis for more proactive environmental management [87,88]. The assessment of freshwater use in LCA needs to better acknowledge freshwater as being both a natural resource and a dynamic compartment in nature, and thereby broaden its scope to include not only impacts from consumptive use, but also in-stream and degradative use. For bio-based production systems, more focus should be on how different forms of water use may affect the regenerative capacity of the system. This requires the development of new impact models that include not only effects on human water needs, but also the water needs of surrounding ecosystems.

### 3.3.3. Biodiversity and ESs

Despite ambitious international and national targets for species conservation and habitat protection (e.g., the Strategic Plan for Biodiversity 2011–2020, following the Convention on Biological Diversity [89], and the Swedish environmental quality objectives [90]), biodiversity losses and ecosystem degradation continue in large parts of the world [45]. In the day-to-day debate, and in most assessment studies, the term biodiversity refers primarily to species diversity (the number of different species in a given area), and impact on biodiversity is measured as the change in species diversity resulting from a studied activity [45,91]. However, the term biodiversity denotes other features besides species level that are less often considered in the assessment, e.g., functional diversity (the function provided by a species or a combination of species in an area), genetic diversity (the genetic variation within a population), ecosystem diversity (the variety of different ecosystems within an area), etc. [45,92]. Additionally, there are qualitative aspects assigned to biodiversity (e.g., conservation targets, conservation status, species abundance, etc.) that are not fully considered when biodiversity impact assessment is limited to changes in species diversity [92]. These different dimensions of biodiversity are constantly interacting in ways that are not well understood. For instance, species diversity and composition influence functional diversity, and genetic diversity impacts on ecosystem dynamics [70].

The functions and ESs (including provisioning ESs, regulation and maintenance ESs, and cultural ESs [89]) provided by biodiversity through its different dimensions play a central role in the

bioeconomy—both in terms of ecological sustainability and the intrinsic value of biodiversity [93], and also in ensuring high productivity, maintaining regenerative capacity, and ensuring the resilience of bio-based production systems. The most obvious link is seen when treating biodiversity as a resource, and an integral part of our natural capital [94]. Over-utilizing a species, causing its extinction, results in a loss in biodiversity, and subsequently a biotic resource that was previously utilized (or had the potential for future utilization) is no longer available for human use (losses in crop diversity and the extinction of commercial fish stocks being notable examples [95,96]). In effect, this means that the resource base of the bioeconomy is eroded, and the potential for bioeconomic growth is diminished. However, there are also other, subtler, ways in which biodiversity affects the regeneration, and long-term productivity, of bio-based production. For instance, in agriculture, biodiversity influences productivity and the rate of regeneration through pollination ESs. A higher diversity of wild pollinators can contribute to higher crop yields [97]. The richness and diversity of pollinators is furthermore affected by ecosystem diversity, where a more mixed and heterogenous landscape provides habitats for a greater diversity of pollinating insects, compared to a more homogenous, monoculture dominated, landscape [98]. Case studies have even shown that increasing ecosystem diversity, by preserving forest habitats as part of the agriculture landscape, can boost pollinator diversity, and improve crop productivity and farmers revenue by as much as 29% for smallholder farms in Tanzania [98]. In forestry, the effects of biodiversity on productivity have also been extensively studied, both in looking at the species diversity–productivity relationship, and also at the effects of forest structural diversity on stand productivity. Results from forestry suggest that both higher species diversity and structural diversity may increase production [99,100], and in fisheries, studies have shown that boosting population diversity can make the production system more stable and resilient to external disturbances [63]. Case studies on salmon fisheries even suggest that boosting population diversity can reduce the frequency unintended fisheries closures due to population collapse by up to ten times compared to a scenario with very low population diversity [63]. In other words, species and structural diversity can increase system resilience, strengthen ESs, and improve the regenerative capacity of the production system [99,101]. In agriculture, these positive effects of diversity on system stability is part of the reason for the growing interest in enhancing the crop genetic diversity in order to improve the climate resilience of agriculture systems [95]. On the soil level, recent studies have shown that soil microbial biodiversity (including species, functional and genetic diversity) plays a prominent role in governing plant productivity, supporting soil formation and nutrient cycling, improving plant resource-use efficiency, and enhancing plant stress resilience. More alarmingly, modern intensive management practices, of these systems, tend to reduce soil microbial diversity, thereby contributing to long-term erosion of system productivity [102].

Thus far, we have concluded that the dominant, and largely unidimensional, approach of measuring biodiversity, and changes in biodiversity, is too simplistic for assessing biodiversity impacts. We have also concluded that this approach can hide some of the environmental consequences of an activity and endanger the long-term sustainability of the studied system. For example, when assessing biodiversity with only a species diversity focus, losses in genetic diversity due to overharvesting can be masked by maintained levels of species diversity. Similarly, if endemic species are replaced by non-native species, the species diversity is unchanged, but impacts on the ecosystem, and the functions provided by the ecosystem, can be significant [103]. To truly assess the sustainability of bio-based production systems, and to avoid unintended negative impacts caused by reductions in one or more biodiversity dimensions, a significantly larger spectrum of biodiversity needs to be considered, rather than what is typically the case. This will require the identification of representative biodiversity indicators for the different dimensions, standardized methods of measuring these, and the development of new models for impact assessment [45,70,104]. Although multiple biodiversity indicators already exist (see for example www.bipindicators.net), the different dimensions of biodiversity are unevenly covered. Most indicators focus on species diversity, followed by ecosystem diversity, whereas genetic diversity and qualitative aspects of biodiversity are underrepresented [45]. Further, many existing indicators are

only applicable to specific geographical regions, or they are calculated based on very location-specific indicator species. On the one hand, this potentially allows for very accurate assessments but, on the other hand, the indicators cannot easily be generalized and applied to areas outside their original region. The development of biodiversity indicators with a cosmopolitan representation, or indicators that can be adapted to the location of the assessment, is much needed [45].

Alongside the development of indicators for biodiversity and ESs, there is also a need to improve the impact assessment models, linking human activities to their effects on the studied indicators. The challenge here is that, for many ecosystems, these impact pathways are not fully understood and/or the data requirements for them to be calculated is not available at a sufficient spatial or temporal precision [45]. According to Chaplin-Kramer et al. [105], the most commonly used LCA methodologies lack the contextual resolution and detailed ecological information required to model the impact pathways from activity to biodiversity impact. In terms of ESs, most guidelines on LCA (e.g., ISO 14040 and 14044, and the International Reference Life Cycle Data System (ILCD) guidelines [1,106]) do not incorporate the ES concept, and the few studies that have done so predominantly focus on provisioning ESs, whereas regulation and maintenance ESs, and cultural ESs, are much less documented [41]. There is a clear risk that this imbalanced coverage of ESs will result in biased interpretations of LCA results, as some services are not covered in the assessment process [41].

One important obstacle to assessing functional biodiversity and ES impacts in LCA lies in the LCA methodology's limited capacity to manage multifunctionality and nonlinear relationships [70,105]. Many ESs are facilitated by the combined effects of multiple ecosystem functions, and their response to stress is often characterized by threshold and nonlinear behavior. This multifunctionality and, at times, counterintuitive behavior is hard to capture with conventional assessment methods [41,107]. For example, the ESs of soil formation are of great importance for the longevity of terrestrial bio-based production, and they are driven by several different processes (e.g., decay of organic matter and mineral weathering), and also feeds into a number of other ESs (primary production, mediation of flows, mediation of biota, etc.) [108]. The different drivers involved in providing ESs, and the interactions, interdependencies, and nonlinear relationships among these processes, are not fully understood, but are highly relevant when assessing the environmental impacts of bio-based production systems. Building this understanding will require research focused on entangling the cause–effect chains linking human activities to environmental impacts, and how these impacts affect ES values and services [41]. To this end, Teillard et al. [47] suggest the increased use of data, models, and modelling methods from the field of ecological science. Othoniel et al. [41] also stress the importance of increased multi- and interdisciplinary approaches to improve the impact assessment models of ESs in LCA. Focusing on improving the spatial and temporal resolution of these models is particularly important, as ESs are often the context-specific sum of multiple functions provided by the specific ecosystem [109]. Since ecosystems, and ecosystem composition, are dynamic and constantly evolving over time, the assessment of their response to impact need to take these dynamics into account as well [110]. For example, the two ESs flood regulation and climate regulation are strongly dependent on the dominating land cover in the studied area. In simple terms, grasslands tend to have a high flood regulation potential but modest climate regulation potential, forests tend to have a higher climate regulation potential compared to grasslands, and croplands, in general, have limited potential for both flood and climate regulation [111]. However, on a landscape level, the transition from one land cover type to another is rarely instantaneous but is rather a transitional process that gradually changes ES potential. Additionally, the potential of the different ESs is also influenced by the surrounding land cover matrix and thus, assessing the long-term impacts on ESs from changes in land cover requires that these interactions are also considered, accounting for the constant evolution of the landscape.

In general, the assessment of impacts on biodiversity and ESs in LCA needs better coverage. Both biodiversity and ESs are complex concepts, involving multiple dimensions and services provided to society. Thus, aggregating these into single impact categories, quantified by a limited set of indicators and characterization factors (as is standard in LCA methodology), may be an oversimplification of

reality, with questionable value for environmental management [41]. Instead, indicators for multiple biodiversity dimensions and ESs are needed, as well as a better understanding of how these dimensions influence the long-term sustainability of bio-based production systems. It is of special importance to enhance the understanding of how human activities impact the genetic and functional dimensions of biodiversity, and how to use this knowledge to design bio-based production systems that support the necessary ESs for long-term productivity and overall sustainability. To achieve this, researchers have suggested closer collaboration with other disciplines, e.g., ecology, systems thinking, and ecosystem science, to improve the impact models, and to better account for the multifunctionality associated with several ESs [13,41,47,112].

## 4. Concluding Discussion

We have shown above that for bio-based production systems to be sustainable, they need to be managed so that the extraction of the resources they provide does not exceed the regeneration rate of the system, and the regenerative capacity of the system must not be diminished by the processes of production, extraction and resource utilization, or by other external factors. The regenerative capacity of the systems is often governed by interactions, and through interdependencies, with surrounding supporting systems. These supporting systems are involved in maintaining the necessary conditions, and providing the critical resources, for regeneration. With these requirements in mind, sustainability assessments of bio-based production systems require a perspective and scope where both the states of the primary system and supporting systems are accounted for. This means that impact categories need to be chosen so that effects on both the production system and the supporting systems are covered. In this study, we have focused on the impact categories "biotic resource depletion", "freshwater use", and impacts on "biodiversity and ESs", as these are particularly important aspects for many bio-based production systems and, also, these constitute aspects that have historically been underrepresented in sustainability assessment studies.

Based on the results of our assessment, we believe there is a need to develop and adapt existing sustainability assessment methods and frameworks to the requirements of bio-based production systems. We also believe there is a need to improve the capacity of the assessment methods to better account for the specific features of impact categories typical for bio-based production systems. Development in this direction poses challenges for several reasons.

First, as described above, the studied impact categories are characterized by being broad, multidimensional concepts but, in most assessments, these multiple dimensions are not accounted for. Instead, a simplistic approach is often taken, focusing on a single dimension or on highly aggregated indicators. To truly assess the environmental sustainability of bio-based production systems, these impact categories need to be disaggregated more, and assessed in their different dimensions and sub-categories. Only at a more disaggregated level can the multiple impact pathways between the impact category and the production and supporting systems be captured in the assessment. What level of disaggregation can be considered "enough" is a question beyond the scope of this review, but it probably depends on how many dimensions are represented in the target system, and the time and resources available to the study. If one accepts the need for disaggregation, this calls for an expanded, and a more detailed, set of indicators than what is often used in sustainability assessments of these systems [45,70,88]. Some indicators are more important for assessing direct impacts (e.g., amount of fish harvested from a fish stock directly affects the stock size), and other indicators are more important for assessing indirect impacts related to the regenerative capacity of the system (e.g., effects of reductions in genetic diversity on regeneration rate in fisheries). Currently, the latter type of indirect impact pathways is poorly covered, partly because it often involves impact pathways characterized by a high degree of nonlinearity, e.g., a sudden collapse of ecosystem functions when the ecosystem is pushed beyond a certain threshold/tipping point [70,88,113].

Covering impacts on both the primary production system and the supporting systems calls for an expansion of the system boundaries of the assessment beyond normal practice. Adopting such a broad

system perspective is a challenge in terms of resource requirements, and also in terms of modelling capacity, data availability, and the often-limited conceptual understanding of several of the impact pathways governing the sustainability of bio-based production systems [70,113]. To better capture these impact pathways in the assessment requires targeted efforts towards several methodological challenges.

## 4.1. Accounting for Spatial Variation

More efforts need to be directed towards improving the level of spatial detail in the assessment, and how variations in geology, topography, land cover and other physical geographical features (also referred to as spatial variations [12]) affect the sustainability of bio-based production systems [12]. For instance, biodiversity and many ESs depend on local geography and landscape configuration [105]. Impacts on biodiversity and ESs from agriculture expansion have, for example, been shown to be strongly affected by the configuration of the surrounding landscape, as this influences the availability of suitable habitats for biodiversity and natural pollinators. Spatial variations in the surrounding landscape also affect local hydrology and cause potential soil erosion [105]. As agricultural systems are heavily dependent on these factors for their regeneration, and also heavily influence them, local geography and spatial variability clearly need to be accounted for in the sustainability assessment of these and other bio-based systems [26].

Increasing the level of spatial differentiation in the assessment requires location-explicit data, and the development of geographically tailored impact models [46]. Typically, however, the necessary data to do this at the local and sub-regional scale are missing, and researchers are left to rely on the extrapolation of national or regional averages [40,51,114]. This can strongly reduce the representativeness of the results of the assessment, especially if the input values are based on national averages for a large and heterogeneous country, encompassing large variations in geology, topography, land cover etc. In such a case, assuming the impact pathways will be homogenous for the entire reference area might be an oversimplification, as these can vary significantly with spatial variations and geographical heterogeneity [14,115].

Accounting for spatial variability to a greater extent than is the case today is likely to be of great importance for ensuring the sustainable development of the growing bioeconomy. The ongoing transformation towards bio-based production, and the increasing use of biomass, is likely to lead to an economy where feedstock is produced, sourced, and processed in a variety of geographically diverse locations—even more so than in the case of current fossil supply chains [116,117]. Transformation towards a more distributed and regionalized, or even localized, economy entails an increasing degree of spatial variation across production systems that needs to be accounted for, in order to ensure that the sustainability of these systems is reliably assessed [9]. In order to achieve this, several researchers have proposed approaches where multiple assessment methods, or features from different assessment approaches, are used in combination [13,46,112,118]. For example, Jeswani et al. [118], and Ford et al. [39] suggested using LCA in combination with Environmental Impact Assessment (EIA), as the EIA approach is designed to take local geography, and potential background pressures that are typically not considered in LCA, into account. The regionalization of LCA, using Geographical Information Systems (GIS), is another promising approach under development [46]. For example, the LCA software OpenLCA [119] allows the use of GIS data for location-specific inventory development, and the defining of regionalized impact factors in the assessment process [120]. However, challenges remain, both in the collection and availability of spatially explicit data, and in how to incorporate necessary spatial features, such as landscape configuration, into the impact models. These are important aspects for ensuring meaningful impact assessments at the regional and sub-regional scale [105], and require an increased use of primary and secondary local data (e.g., with the help of GIS and satellite and image analysis technologies), and for an increased use of local knowledge that would be integrated into the impact models [13,105]. Development in this direction could increase the accuracy of the assessment, but also require more time and effort devoted to data collection, processing, and analysis, as well as increasing the level of complexity of the assessment.



### 4.2. Local Environmental Uniqueness

Closely related to spatial variations are what Reap et al. [12] describes as features of "local environmental uniqueness". This denotes non-physical, spatially varying, parameters and characteristics of a system that influence how sensitive the system is to external pressures. Examples include soil quality factors, soil buffering capacity, population density, etc. [12], but environmental uniqueness can also refer to qualitative aspects, such as the type of farming practices used, or variations in qualitative and genetic aspects of biodiversity (e.g., occurrence of endemic or red-listed species) [12,41]. These factors can strongly influence the sustainability of a production system and its environmental impacts. For bio-based production systems, these, often intangible, factors can be particularly important to consider in the assessment, as they influence the production system, its surrounding supporting systems, and the shape and magnitude of interlinkages between the two [13,26]. For example, soil quality properties, such as soil organic carbon (SOC) content, water holding capacity, texture, chemistry, microbiology, etc. are examples of local environmental uniqueness of great importance for soil productivity and resilience [121,122]. In an agricultural production system, productivity and environmental impact are both strongly affected by these soil quality parameters [41]. High-quality soils can give greater yields per unit effort than low-quality soils, and lower quality soils may require more intensive farming practices to be economically productive, e.g., in the form of additional fertilizer use and intensified tillage, increased nutrient runoff, and subsequent environmental impacts [123]. Since soil quality can vary significantly within and between regions, this aspect of local environmental uniqueness can strongly influence farming practices and, subsequently, the sustainability of seemingly very similar production systems [26].

It is also important to keep in mind that many aspects of local environmental uniqueness are not static but are rather highly dynamic parameters that are continuously changing in response to external pressures. For instance, SOC contributes to several beneficial soil functions, including soil productivity, carbon sequestration, and water and nutrient retention [124]. Soil tillage practices can also increase agricultural productivity by improving soil structure. However, long term, intensive tillage can also cause the depletion of SOC by accelerating decomposition [124]. With losses of SOC beyond a certain threshold, the benefits previously provided by SOC start to decline, and soil productivity is eroded, further increasing the need for an intensification of tillage and other farming practices, in order to maintain productivity. The result is a reinforcing feedback loop of decreasing soil quality, leading to reductions in soil productivity. This simple example illustrates that soil quality, and other parameters of environmental uniqueness, cannot be treated as spatial and/or temporal constants in the sustainability assessment of these systems. At a high level of SOC, or any other parameter of environmental uniqueness, the environmental impacts from a production system (e.g., tillage farming) may be negligible, or even beneficial, for productivity. From a long-term perspective, however, if the disturbance continues, and the parameter decreases beyond a certain level, this can trigger feedback loops causing the environmental impacts of the previously sustainable production system to escalate.

Most methods for environmental assessment, including state-of-the-art LCA, tend not to capture many important aspects of local environmental uniqueness. At best, a simple characterization is made where the dimensions of environmental uniqueness are organized into discrete categories (e.g., farming practices are commonly categorized into conventional vs. organic [26]), but, in many cases, such differentiation is non-existent (e.g., in biodiversity assessments, species vulnerability or endemism are rarely accounted for [43]). The reality is that dimensions of environmental uniqueness cannot be treated as discrete categories, but should rather be seen as a range, or spectra [26]. The studied system is constantly moving along these spectra, shaped by synergies and feedback interactions triggered by its own management, and by influences from surrounding systems [26].

Accounting for environmental uniqueness, and for the interactions and emergent system properties these create (feedback relationships, system thresholds, etc.), is a significant challenge that needs to be addressed in order to further improve the sustainability assessment of bio-based production systems [112,113,125]. While significant research is being conducted on developing more regionalized,

or even location specific, environmental assessments [126,127], the limited availability of impact models accounting for the unique features of the local environment remains an obstacle. Instead, practitioners are often forced to rely on more readily available, site-generic impact models, due to the lack of detailed knowledge regarding the local uniqueness of the territory studied [15,26,126,128]. Adding the necessary level of detail to the assessment will often require extensive data collection, and close collaboration with local stakeholders and experts from multiple disciplines and sectors [112,126,129]. This approach, building on local knowledge as a central component in the assessment and management planning, has been advocated in several studies, including case studies on coastal and freshwater fisheries in Sweden, the Mediterranean, Brazil and Southeast Asia [130–132], forest biodiversity management in Europe [133], and on sustainable agriculture development in the UK [134]. Furthermore, examples of collaborations across scientific disciplines and methods to capture and analyze this environmental uniqueness have been documented in studies where, e.g., LCA methods have been combined with GIS [120,126], System Dynamics modelling [88], Ecosystem modelling [105], and Group Model Building [135].

There is a risk that important environmental impacts may be overlooked if environmental uniqueness is not considered in the assessment [92,105]. However, adding levels of detail to the assessment exemplified above is resource intensive, and requires a greater degree of cross-discipline collaboration and stakeholder involvement than what is common in conventional assessment methods [105].

## 4.3. Environmental Dynamics

The importance of accounting for environmental dynamics has been introduced in Section 4.2., exemplified with the change in the decomposition rate of SOC in response to tillage. More generally, environmental dynamics refer to temporally changing aspects that influence the state of the studied system, its interactions with surrounding systems, and the magnitude of its environmental impact. This includes, for example, the timing and rate of release of emissions, timing and rate of resource extraction, temporal delays, seasonal variations, etc. [12]. Bio-based production systems are particularly sensitive to these factors, and they can significantly influence their environmental performance. For instance, seasonal food web dynamics can significantly affect biodiversity impacts from fish harvesting [9], and provisioning of ESs can be more or less affected by a stressor depending on its timing over the year [41,107]. There can be temporal delays between the time of environmental impact and the observed effect on ESs (e.g., due to variations in supply and demand for the service over time [107]), and the productivity and input requirements of a production systems can change with its age (e.g., perennial cropping systems exhibit nonlinear patterns of increasing and then decreasing yields per unit effort over their lifetime [136]).

Environmental dynamics can also indirectly influence the long-term sustainability of bio-based production systems via their effects on key supporting systems. The hydrological system is one example. As described above, freshwater is a critical resource for most terrestrial bio-based systems, and in many areas, groundwater is the dominating source [137]. Groundwater availability is determined by the local hydrology, influenced by several factors, such as percolation rate, soil type, temperature, etc. However, the hydrological system is also characterized by temporal and spatial delays, meaning that it takes time before the full effect of a disturbance at one part of the system is experienced in other parts of the system [138]. For instance, a farmer may want to increase productivity by investing in irrigation. Therefore, the farmer decides to increase groundwater use by drilling a new well and increase extraction. Due to the temporal delays in the hydrological system, an immediate effect on the groundwater level in neighboring wells may not be experienced. Depending on the local hydrology, it may take one, or several, years before the increased water extraction affects neighboring wells further downstream. Once the effect has reached neighboring areas, and the farmer cuts extraction back to its original value, the decline in groundwater level will continue for some time before it slowly starts increasing back to its original level [138]. The dynamics of the hydrological system makes sustainability

assessment and management of any bio-based production system relying on the water resource very difficult, as the true effect of altered water use is only seen after a considerable time delay. If, during this delay, the farmer has made capital investments in an irrigation system, and expanded irrigation to finance these investments, he may now find himself in a lock-in, relying on an unsustainable exploitation of water resources in order to maintain productivity and economic profitability.

Environmental dynamics are often overlooked in environmental assessment studies. In LCA, a steady-state approach is typically taken, where emissions and resource consumption occurring throughout the studied lifecycle are aggregated into a single value and assigned to a given point in time. The potential environmental impacts are characterized using predominantly linear impact assessment models and thus, time-dependent changes in environmental processes, in the production system, or in the environment responsiveness to stress, are not considered [139,140]. This can be problematic for the reliability of the assessment, as environmental stressors are often stochastically spread out in time, and the magnitude of their impact fluctuates with the development of the receiving system, and with the accumulation of environmental pressure [10,26,88,104,139]. This approach reduces the reliability of the assessment as a tool for policy development and scenario-based planning. With growing interest in using LCA for predictive modelling, bioeconomy strategic planning, and the assessment of emerging technologies, several authors stress the need to either develop the LCA methodology, or to complement it with other methods in order to better integrate temporal dynamics [10,13,47,112,141,142]. A challenge, though, is that most operational LCA methods do not incorporate the necessary temporal information, or the required case-specific data are not available, to account for environmental dynamics in the impact assessment [41,140]. On the one hand, deepening the assessment to include this information could result in a substantial level of detail and an increase of depth to the assessment. On the other hand, Almeida et al. [143] stresses that it is not always the case that the extra effort required to increase the temporal resolution of the LCA matches the potential gain in results. Whether or not this is the case depends on the context and objectives of the study. It is likely that short-lived processes are less affected by temporal dynamics than long-lived processes, and it is also likely that sensitivity to temporal dynamics varies between different systems [104,143].

In conclusion, adding temporal dynamics to the assessment of bio-based production systems could, in many cases, allow for a more accurate impact assessment. These systems are typically strongly affected by seasonal variations, and the production system and its supporting systems are constantly evolving, causing changes in their environmental performance and response to environmental stressors. Unless these dynamics are taken into consideration in the assessment process, the results, and the following policy actions, are likely to be based on a static system perspective when, in fact, the system is highly dynamic. However, with the added effort this type of assessment might entail, it is up to the practitioner executing the assessment to evaluate how significant environmental dynamics are to the objectives of the study, and for which aspects of the impact model it is worthwhile to invest in temporal differentiation [104,144].

## 5. The Way Forward

### 5.1. Expanding System Boundaries

Ensuring the sustainability of bio-based production systems requires assessments methods that are tailored to the specific characteristics of these systems. To ensure long-term sustainability, the rate of resource extraction must not exceed the rate of regeneration, and the regeneration capacity of the system must not be diminished [24]. In an environmental sustainability assessment, this entails expanding the system boundaries beyond conventional practice so that both the primary system and its supporting systems are covered, and the cross-system interactions involved in providing the necessary conditions for regeneration are accounted for [23]. "What are the conditions necessary for maintaining the regeneration capacity of the systems?" and, "what are the supporting systems

necessary for maintaining these conditions?" are two helpful questions for practitioners to consider when deciding upon the system boundaries at an early stage of an assessment study.

## 5.2. Rethinking Impact Categories

Broader system boundaries call for a subsequent expansion of the environmental impact categories covered, so that impacts on both the primary system and its supporting systems are considered. More specifically, efforts must be targeted towards improving the coverage, and impact models, of categories affecting the regeneration capacity of bio-based systems. In this study, we have focused on a select few, historically underrepresented, impact categories, that are generally agreed to be of great importance for the long-term sustainability of many bio-based production systems (biotic resource depletion, water use and biodiversity and ESs).

These impact categories are challenging to assess because they are characterized by being multidimensional and complex concepts, often with multiple factors affecting the primary production system and surrounding supporting systems. In most assessments, however, these impact categories are represented in a highly aggregated format. In order to truly model these impact categories, they need to be disaggregated into their different dimensions, accompanied by suitable indicators for each dimension. For example, biodiversity needs to be differentiated into dimensions including species diversity, genetic diversity and functional diversity, each accompanied by representative indicators to measure their impacts on both the primary production system and its supporting systems.

## 5.3. Contextualizing the Impact Models

Once sufficiently broad system boundaries have been set (including both the primary system and its supporting systems), key impact categories have been identified, and their different dimensions and indicators established, the impact pathways connecting the impact categories with the bio-based production systems need to be understood and modelled. Entangling these impact pathways requires targeted efforts towards moving away from generalized assessments, and instead moving towards more context specific impact models. By increasing the level of spatial- and temporal differentiation in the assessment, including more details on geographical variations, environmental uniqueness, and environmental dynamics of the system, a more representative assessment of its long-term sustainability can be achieved. This entails several methodological challenges, some of which were discussed in Sections 4.1–4.3. To overcome these challenges, we advocate increased collaboration with other research fields, such as ecological science, system theory, risk modelling, scenario analysis, etc., increased collaboration with local stakeholders and actors with local ecological knowledge, and the use of approaches where multiple modelling methods are applied in combination (e.g., LCA is combined with GIS, ecological modelling methods, and System Dynamics modelling).

Before initializing this type of cross-discipline and multi-modelling approach, one should be aware that the efforts and resources required for such an assessment can be significant. If reliable location-specific data and impact models are not available, these need to be collected and developed as part of the assessment process. With the inherent complexity of many bio-based systems, and the often incomplete understanding of the processes and feedback loops connecting them to their supporting systems, making assumptions regarding the structure and dynamics of the impact pathways may be required in the modelling process. Some would argue that expanding the assessment as advocated above will only lead to added uncertainty and costs, while being of little help in guiding the transition to a sustainable bioeconomy. We argue differently. As presented in this review, taking a systems approach in sustainability assessments, tailoring it to the context of the study by including a significant degree of case-specific information and impact models, is a necessity to ensure the criteria for the sustainability of these systems are met. It is true that the models developed for such an assessment will never constitute perfect representations of reality, and a certain degree of uncertainty is unavoidable. However, the complex nature of bio-based systems cannot be ignored. Even an assessment based on imperfect models, as long as it is built on best-available knowledge and transparent assumptions,

is likely to be better guidance for sustainable development than an approach where the complexity of the issue is marginalized, and significant drivers of environmental degradation are intentionally left out due to limitations in modelling capacity.

**Funding:** This research received no external funding.

**Acknowledgments:** We would like to acknowledge Liisa Fransson from RISE Research Institutes of Sweden for providing valuable input and supervision throughout the review and writing process.

**Conflicts of Interest:** The author declares no conflict of interest.

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
