# Peer review of "Bio-Based Production Systems: Why Environmental Assessment Needs to Include Supporting Systems"

_sustainability, doi:10.3390/su11174678_

Round 1

Reviewer 1 Report

This is a valuable and well written contribution of how LCA studies should extend to the supporting systems of biomass production. The authors conduct a literature review to elucidate the shortcomings of current LCA studies regarding biomass production systems. There are some weaknesses regarding precision and transparency of the analysis. I recommend to publish this article after addressing the following issues:

It is not clear what subsumed under bio-based production systems. Please provide a definition as well as examples for the most common ones and what they mainly produce or in which value chains the products feed into (e.g. depict this information using a table). Thereby you can increase the relevance of the article as other scholars then can identify important impact categories for their value chains.

The term “sustainability assessment” or “sustainable” in this manuscript refers to only 1 pillar of sustainability, that is the environmental sustainability. What about the other pillars of sustainability? For example, land-use change is related to substantial social and economic impacts. To avoid this question, specify or change the term “sustainability assessment”. Unless the other two pillars are covered in the data collection and the analysis, these must be integrated in the results and concluding remarks.

Research question 1 could be more precise: What do you mean with “complicating factors” exactly? From reading the manuscript I know it refers to differences in comparison to fossil-based systems. However, the question could be formulated more precise.

Line 198: Note that also fossil resource systems are not “inert” (e.g. pumping oil from a reservoir can lead to soil compaction). Generally, I do not agree with this term coming from chemistry referring to a substance that is not chemically reactive.

Chapter 2:

In this context: How did you make sure that the key words are appropriate to find articles related to bio-based production systems? This question is related to the previous comments on the term “sustainability assessment” and “bio-based production system”.

Why the time span 2008-2018? Is there any reasoning behind it?

It is not clear how you analyzed the articles. Please provide a more detailed description about the conceptual background. For example in line 173 you should define at this point what counts as “critical”. Maybe the chapters 3.1-3.3 could be considered as the conceptual background for your analysis, by which allows you to identify and describe the four impact categories.

Chapter 3:

It is not clear to me why you chose the four impact categories biotic resource depletion, freshwater use, biodiversity loss and impacts on ES.

Chapters under 3.3: Some numbers regarding the impact could be provided to show how supporting systems are effected

In general, there are very long sentences e.g. line 816-822. Try to size them down to make the manuscript easier to read.

Line 427: term LCIA used without explanation

Chapter 5: Why did you not include any new references? Since this is a discussion you should add literature.

Author Response

Thank you for your valuable comments. Please see response below.

Comments and Suggestions for Authors

This is a valuable and well written contribution of how LCA studies should extend to the supporting systems of biomass production. The authors conduct a literature review to elucidate the shortcomings of current LCA studies regarding biomass production systems. There are some weaknesses regarding precision and transparency of the analysis. I recommend to publish this article after addressing the following issues:

It is not clear what subsumed under bio-based production systems. Please provide a definition as well as examples for the most common ones and what they mainly produce or in which value chains the products feed into (e.g. depict this information using a table). Thereby you can increase the relevance of the article as other scholars then can identify important impact categories for their value chains.

Thank you for this comment. We agree to your suggestion and have added a definition of bio-based production systems, including some examples. Pleas see line 160 - 163. However, we decided not to add a table of system categories as this could give the impression that systems of the same category all can be managed in the same way. The diversity of systems, and considerable variation even within systems of the same category, calls for approaches that are tailored to the specific characteristics of the studied system.

The term “sustainability assessment” or “sustainable” in this manuscript refers to only 1 pillar of sustainability, that is the environmental sustainability. What about the other pillars of sustainability? For example, land-use change is related to substantial social and economic impacts. To avoid this question, specify or change the term “sustainability assessment”. Unless the other two pillars are covered in the data collection and the analysis, these must be integrated in the results and concluding remarks.

The study has focused on the environmental pillar of sustainability. Please see the added clarification in line 62 - 67.

Research question 1 could be more precise: What do you mean with “complicating factors” exactly? From reading the manuscript I know it refers to differences in comparison to fossil-based systems. However, the question could be formulated more precise.

Thank you for this suggestion. The question has now been clarified and adjusted. Please see line 86 - 88. Adjusted to: (1) “What are the key characteristics of bio-based production systems that need to be taken into account when assessing their long-term environmental sustainability?”

Line 198: Note that also fossil resource systems are not “inert” (e.g. pumping oil from a reservoir can lead to soil compaction). Generally, I do not agree with this term coming from chemistry referring to a substance that is not chemically reactive.

By “systemically inert” we refer to whether or not the resource system is interacting with its surrounding systems or not. True that oil extraction can lead to soil compaction, but this is a result of the extraction process not a consequence of the state and size of the resource stock because the oil stock. However, we agree to your comment that “inert” is inappropriate wording as there can be interactions we are not aware of. Please see adjustment in line 206.

Chapter 2:

In this context: How did you make sure that the key words are appropriate to find articles related to bio-based production systems? This question is related to the previous comments on the term “sustainability assessment” and “bio-based production system”.

As explained in line 115 – 118. We intentionally applied a broad search strategy in order not to exclude relevant studies not explicitly using the terminology LCA, environmental assessment, bio-based production systems, etc. Articles outside the scope of the study were filtered out based on the selection criteria presented in line 121 - 125.

Why the time span 2008-2018? Is there any reasoning behind it?

The 10 year time span was set to limit the scope and thereby make the literature study manageable. With the growing interest in the bioeconomy and bio-based economy that has been witnessed over the last decade or so we believe most relevant studies on the topic will be covered even with this limited time span.

It is not clear how you analyzed the articles. Please provide a more detailed description about the conceptual background. For example in line 173 you should define at this point what counts as “critical”. Maybe the chapters 3.1-3.3 could be considered as the conceptual background for your analysis, by which allows you to identify and describe the four impact categories.

Thank you for this very valuable comment. Please see adjustments made in chapter 2, line 133 – 147.

Chapter 3:

It is not clear to me why you chose the four impact categories biotic resource depletion, freshwater use, biodiversity loss and impacts on ES.

Our analysis suggests that there are four impact categories that are particularly important for the sustainability of bio-based production systems. Also they are among the least represented impact categories in LCA studies. Please see clarification in chapter 3.2. and line 294 – 298.

Chapters under 3.3: Some numbers regarding the impact could be provided to show how supporting systems are effected

Thank you for this suggestion. Some additional numbers and data have been added to the argumentation where relevant (see for example line 375 - 376 & 566 – 569). However, we want to avoid too many case examples as these might not be representative at the conceptual level of the paper.

In general, there are very long sentences e.g. line 816-822. Try to size them down to make the manuscript easier to read.

The manuscript has been revised once more for readability.

Line 427: term LCIA used without explanation

Thank you for this observation. It has now been adjusted (see line 349).

Chapter 5: Why did you not include any new references? Since this is a discussion you should add literature.

Chapter 5 should not be seen as a discussion, but rather as recommendations for future efforts to improve the sustainability assessment and management of bio-based production systems. These are the conclusions and reflections of the authors, based on the key findings and concluding discussion presented in the manuscript. This aside, we have added a two important references that are key to the conceptual background behind the reasoning on sustainability (Goodland and Daly, 1996) and bio-based production systems (Lan Ge et al., 2016).

Reviewer 2 Report

A very relevant paper on the assessment of biobased production systems. It is well describing the important issues!

Author Response

Thank you for this very nice comment and for taking time reviewing our work.

Reviewer 3 Report

Thank you for your well written paper. It was enjoyable to read and I found the literature review beneficial.

Author Response

Thank you for this very kind comment and thank you for taking time reviewing our work.

This manuscript is a resubmission of an earlier submission. The following is a list of the peer review reports and author responses from that submission.

Round 1

Reviewer 1 Report

Dear authors,

Thank you for your very interesting and informative manuscript. The issue of LCA and bioeconomy is currently a highly discussed topic. The manuscript is overall well written & structured, the research approach is transparent, your approach is comprehensible, etc. Nevertheless, there are some issues which in my opinion weaken your manuscript. Please find below some comments and questions which should help to strengthen the focus and significance of your paper:

In my opinion, there is a lot of (‘general LCA’) information I can find in almost every LCA handbook. I would recommend highlighting the specific issues for bioeconomy/bio-based systems/food systems.

You should focus more (especially in section 4) on the actual problem of LCA in the context of bioeconomy. Delete some of the ‘general information’ (think about who your target audience will be) and strengthen your focus on LCA in bioeconomy.

Please check your aim and objective: L97: Objective: use the same wording as you did for your aim. This will help the reader to follow you and understand your approach “reliability of LCA as an assessment and planning tool for the transition towards a bioeconomy” vs. “reliably and usefulness of LCA as a tool for sustainability assessment and policy development”. Also in your discussion/conclusion try to refer to your research questions. Again this will help the reader to follow.

Why do you restrict your manuscript to food systems? And if so, please strengthen your focus on that, rather than providing examples e.g. for biofuels, etc. I do not see the need for this restriction.

check your citation style (e.g. L245 & L247)

L118: Why did your search terms not include LCA and bioeconomy or LCA and bio-based systems?

L136-170: Seems like a “nice to have” but the information you provide here might be too much. Focus or highlight the connection to bioeconomy.

I understand that your focus is on impact and interpretation, but what about goal and scope definition? To me, it occurs that – at least - functional unit and system boundary selection is in particular interesting in terms of bio-based system especially when compared with fossil based system (-> substitution effect of bioeconomy). The explanation you provide (L56) is – at least for me - not enough. 

Background: Please check this section so that you do not repeat things already stated in the introduction. -> Maybe restructure that part: The information about bio-based system and food systems could be included in the introduction part

3.2 What I would expect is specific literature/background on LCA in the specific context rather than this more general information about LCA. Maybe it is just wording, but please try to strengthen your focus.

Problem areas: In general, there is a lot of very important and interesting information in this section. But there is so much information about the general aspects of these problems, that the actual content of your work is underrepresented. I would highly recommend that you strengthen your results and use the more general aspects only in minor content. Example: 4.1: Most of the things stated here (L237-289) can be found in almost every LCA book/paper/etc. This is not new and for me is also not specifically interesting in the context of bioeconomy.

I do not clearly understand why you choose the three impact categories – just because they are underrepresented? What are to impact categories addressed by other publication in the context of bioeconomy? Would that not be interesting?

Figure 1: I do not see the need for this figure. Again it might weaken your results (considering that this is not your focus)

L780-786: Something like this ‘summary/recommendation’ I would expect for each problem area.

I do miss information if you found additional problem areas besides the ones already defined by Reap at al. (2008)? For instance, you briefly mentioned the need for LCA for emerging technologies. From my experience this is a huge topic in LCA in the context of bioeconomy (-> LCA at the early stage of product/technology development); also who is the driver for LCAs in the context of bioeconomy (-> see for instance any kind of R&D projects funded by the European Union. etc.) You do not have to go into detail but just think of it as additional input.

Reviewer 2 Report

The review article contains a lot of interesting information, it is well written, and the starting point (with Reap et al's article) creates a good framework for discussing the objectives pursued by the authors. The challenge with the article is that it tries to cover too many issues at once and that some of the main concepts are poorly defined (as I will soon return to). As a reader, I would also like to understand what the authors believe the goal of further refinement of LCA in connection to the bioeconomy would be. From the text, it seems like the authors believe that in due time, given that enough effort is put into the task, LCA could perfectly replicate the world. One of the basic, and insolvable, problems of LCA is that there is no possibility to check whether results are correct or not. We do not really know if refinement efforts will pay dividends in the form of more correct assessments or whether we are just adding another false level of security. I do not claim that the article needs to present a metaphysical position on theory of science, but it needs to deal with a clear message to convey. Three things can be done to improve the paper substantially: 1) Better define the specifities of bioeconomy and food production; 2) make clearer connections between the methodological problem areas in LCIA on one hand and bioeconomy and food production on the other; and 3) ending the paper in a fashion where the elements are wrapped up and results are connected back to the initial research questions.

1)     Better define the specifities of bioeconomy and food production

The article repeatedly state that there is something special about bioeconomy and food production. It is full of sentences like “Bio-based systems (both natural and man-made) are known for their inherent complexity and multifunctionality” (lines 76-77), “The complexity of bio-based systems, and the synergies and feedbacks connecting different aspects of sustainability, are factors that complicate the assessment and planning of bioeconomy development” (lines 191-193), and “As has been mentioned several times in previous sections, bio-based systems are characterized by their inherent complexity which, in combination with methodological limitations in impact modelling, adds to the uncertainty of the assessment” (lines 757-759). There are, however, very few examples of what this complexity consists of how and how it differs from systems that are not bio-based. In order to judge whether Reap et al’s article is applicable to/relevant to bioeconomy systems, the reader must be presented with a clear image of what bioeconomy systems look like and preferably juxtaposed with non-bio-based systems.

2)      Make clearer connections between the methodological problem areas in LCIA on one hand and bioeconomy and food production on the other

One of the reasons why it might be difficult to spot the specifities of bioeconomy and food production is because much of the text describing the seven problem areas rather define general problems in LCIA than problems specifically linked to bio-based systems. Some the sections, especially the one on biotic resource depletion, is to the point and connects well with an understanding of something that must clearly be developed to judge the performance of a bio-based system. Other sections contain mostly text on generic problems in LCIA and show that many of the problems identified by Reap et al still exist. A remedy for this could be to free the text from having to describe all of the seven areas in detail and focus on issues that are specific for bio-based systems. In the end, I, as a reader, want to know what is needed in developing LCA for decision making relating to the bioeconomy specifically. I don’t just want the conclusion that LCA is uncertain but if one spent an infinite amount of money and used all clever modelling techniques, one could in fact get a correct answer. I still wouldn’t know if that answer meant that a system was sustainable or not.

3)      Wrap up the paper with conclusions that underpin the connections between the research questions and the results from the literature review

Throughout the paper, solutions to problems are given in the form of “mights” and “coulds”, meaning that methodological developments are presented – and many of them – without having the time and space to show exactly how they will solve the overall problem of using LCA for sustainability assessment and policy development in the bioeconomy transition. The three problem domains (Managing modelling limitations, Transparency in value judgements, and Clarifying uncertainties) defined at the very end of the paper does little to bring together all advices given in previous paragraphs. These problem domains are left hanging and brings in new elements like “these systems always entail a level of uncertainty”, a conclusion that hasn’t been presented earlier. We are not told how exactly the problem domains connect to the preceeding text and what they mean for the application of LCA for decisions about bioeconomy systems.

My proposition is that you select fewer of Reap et al’s problem areas and present only those that are closely connected to challenges exclusive to (or predominantly found in) the bioeconomy. This calls for a better description of the specifities of bioeconomy and food production in the preceeding sections. Lastly, the conclusions should bring these elements together and perhaps clearer state what the pressing issues are if one has to prioritize.

Overall, I want to repeat that the text is well written. All the paragraphs in themselves make sense and are interesting. The article would just gain from removing some parts, adding some others, and through those measures create a more consistent argument.